# A comparison of disseminated intravascular coagulation scoring systems and their performance to predict mortality in sepsis patients: A systematic review and meta-analysis

**Girum Tesfaye Kiya**[1]*, **Gemeda Abebe**[1], **Zeleke Mekonnen**[1], **Edosa Tadasa**[1], **Gedion Milkias**[2], **Elsah Tegene Asefa**[3]

1 School of Medical Laboratory Sciences, Jimma University, Jimma, Ethiopia, 2 Department of Medical Laboratory Science, Arbaminch Health Science College, Arbaminch, Ethiopia, 3 Department of Internal Medicine, Jimma University, Jimma, Ethiopia

* tesfaye.girum@ju.edu.et

**Data Availability Statement:** All relevant data are within the manuscript and its Supporting Information files.

## Abstract

### Background

Disseminated intravascular coagulation (DIC) is a common complication in sepsis patients which exacerbates patient outcomes. The prevalence and outcomes of DIC in sepsis is wide-ranging and highly depends on the severity of the disease and diagnostic approaches utilized. Varied diagnostic criteria of DIC have been developed and their performance in diagnosis and prognosis is not consistent. Therefore, this study aimed to determine the score positivity rate and performance of different DIC scoring systems in predicting mortality in sepsis patients.

### Methods

Four databases, including Medline (through PubMed), Scopus, Embase, and Web of Science were searched for studies that determined DIC in sepsis patients using the three scoring systems namely: the International Society on Thrombosis and Hemostasis DIC (ISTH-DIC) criteria, the Japanese association for acute medicine DIC (JAAM-DIC) criteria, and the sepsis-induced coagulopathy (SIC) criteria. A random-effect meta-analysis was performed with a 95% confidence interval (CI). Subgroup analysis was conducted in view of geographic region and sepsis stages. the protocol was submitted to the Prospective Register for Systematic Reviews (PROSPERO) with an identifier (CRD42023409614).

### Results

Twenty-one studies, published between 2009 and 2024, comprising 9319 sepsis patients were included. The pooled proportion of cases diagnosed as positive using ISTH-DIC criteria, JAAM-DIC criteria, and SIC were 28% (95% CI: 24–34%), 55% (95% CI:42–70%), and

**Funding:** GTK, ET, GM, and ETA are supported by the Research and Innovation Director (RID) office of Institute of Health, Jimma University with Mega Research Fund Scheme (2022-2025). The funders had no role in study design, data collection and analysis, decision to publish, or preparation of the manuscript.

**Competing interests:** The authors have declared that no competing interests exist.

57% (95% CI: 52–78%), respectively. The pooled mortality rates were 44% (95% CI:33–53%), 37% (95% CI: 29–46%), and 35% (95% CI: 29–41%), respectively. The pooled sensitivity and specificity of ISTH-DIC to predict mortality were 0.43 (95% CI: 0.34–0.52), and 0.81 (95% CI: 0.74–0.87), respectively, while for JAAM-DIC it was 0.73 (95% CI: 0.57–0.85) and 0.46 (95% CI: 0.28–0.65), respectively. Pooled sensitivity and specificity for SIC were 0.71 (95% CI: 0.57–0.82) and 0.49 (95% CI: 0.31–0.66), respectively.

## Conclusion

The SIC and JAAM-DIC scores exhibited higher sensitivity to identify patients with coagulopathy and predict patient outcomes, and thus are valuable to identify patients for possible treatment at an early stage. The ISTH-DIC score perhaps identified patients at later stages and demonstrated better specificity to predict disease outcomes. Thus, early identification of patients using the SIC and JAAM-DIC scores and later confirmation using the ISTH-DIC score would be beneficial approach for improved management of patients with sepsis.

## Introduction

Disseminated intravascular coagulation (DIC) is an acquired syndrome characterized by intravascular activation of coagulation with loss of localization arising from different causes. It can originate from and cause damage to the microvasculature, which if sufficiently severe, can produce organ dysfunction [1]. A wide range of diseases are associated with DIC with different corresponding clinical symptoms [2]. DIC is common in sepsis and septic shock patients and is associated with poor prognosis in these patients [3, 4]. Depending on the severity of the disease and the diagnostic criteria used, the prevalence of DIC in sepsis patients ranged from 17% to 61% [4–7]. A crosstalk between inflammation and coagulation [8], increased expression of tissue factor [9], suppression of fibrinolysis [10], and activation of platelets [10] are crucial mechanisms of DIC in sepsis.

The International Society on Thrombosis and Hemostasis (ISTH) established overt DIC diagnostic criteria that apply to DIC diagnosis regardless of the underlying disease [1]. The ISTH-DIC criteria have been used as a global standard though there is no gold standard for diagnosing DIC [10]. The overt DIC scoring system was developed based on the previous scoring criteria called Japanese Ministry Health and Welfare (JMHW) DIC criteria, which comprises clinical data such as the underlying disease, symptoms, and laboratory data such as fibrin/fibrinogen degradation product (FDP), platelet count, fibrinogen, and prothrombin time ratio [11]. The presence of underlying diseases is a prerequisite to using the ISTH DIC criteria while it was part of the score in the previous JMHW criteria. The FDP is also replaced by a fibrin-related marker (FRM), which encompasses soluble fibrin and D-dimer, as FDP is not widely available outside Japan. The reported drawback of ISTH overt DIC criteria is a delay in diagnostic timing, which in turn has implications on disease outcomes [12, 13].

The other commonly used diagnostic criteria is the Japanese association for acute medicine-DIC (JAAM-DIC) criteria, which involve systemic inflammatory response syndrome (SIRS) and eliminates fibrinogen, unlike the previous criteria [14]. Dynamic changes in the platelet count within 24 hours were also included in the JAAM-DIC. The criteria is more sensitive at an early stage as compared to the ISTH DIC criteria [6, 15]. However, the SIRS category

that makes up the JAAM DIC criteria is no longer used in the sepsis-3 definition, necessitating other criteria which best fit with the new definition.

A nationwide retrospective survey in Japan produced new diagnostic criteria for sepsis-induced coagulopathy (SIC), which is based on platelet count, prothrombin time ratio, and sequential organ failure assessment (SOFA) score [16]. The SOFA score is computed from dysfunction of the respiratory, cardiovascular, hepatic, and renal systems whereby a score of 2 or more within each of these systems was defined as organ dysfunction [17].

Previous studies have examined the diagnostic and prognostic performance of the aforementioned scoring systems in sepsis patients. The proportion of DIC in sepsis when applying ISTH-DIC criteria ranged from 16% to 45% [18, 19], while the proportion based on JAAM-DIC criteria ranged from 29% to 91% [20, 21]. Based on the SIC criteria, the proportion of coagulopathy in sepsis patients ranged from 22% to 86% [19, 22]. Similar variability in mortality rate and predictive performance of outcomes have been observed. However, pooled estimate of the proportion of positive scores, mortality rate, and predictive performance of these scores, by making use of meta-analysis in sepsis patients is lacking. This systematic review and meta-analysis assessed the proportion of DIC and coagulopathy in sepsis patients based on the ISTH-DIC, JAAM-DIC, and SIC criteria and evaluated their performance in predicting mortality in sepsis patients.

## Methods

This systematic review and meta-analyses was performed according to the guidelines in the Preferred Reporting Items for Systematic reviews and Meta-Analyses Statement (PRISMA 2020) [23], and the protocol was submitted to the Prospective Register for Systematic Reviews (PROSPERO) with an identifier (CRD42023409614).

### Search strategy

An electronic search of published literature was conducted on March 20, 2023 and updated on August 30, 2024. Four databases, including Medline (through PubMed), Scopus, Embase, and Web of Science were searched. The following search terms were used: 'Disseminated Intravascular Coagulation', 'Disseminated intravascular Clotting', 'DIC', 'coagulopathy', 'Disseminated intravascular coagulopathy', consumptive coagulopathy', ISTH, 'International Society on Thrombosis and Haemostasis', 'Overt DIC' 'Overt disseminated intravascular coagulation', 'Japanese Association for Acute Medicine', JAAM, 'Sepsis-induced coagulopathy', SIC, Score, 'Scoring system', 'Criteria', 'Diagnostic', 'Diagnostic criteria', 'Prognosis', 'performance', 'Outcome', 'Prognostic', 'Predict', 'Mortality', 'Death', 'Fatality', 'Lethality', 'Sepsis', 'Severe sepsis', 'Septic shock', 'systemic inflammatory response syndrome', 'SIRS', 'septicemia', 'septic', 'Blood Poisoning', 'SOFA', 'Sepsis 3'. The detailed search strategy is presented in the S1 Table.

### Inclusion and exclusion criteria

Studies were included based on the following inclusion criteria: 1) studies involved sepsis patients; (2) observational studies; 3) studies that describe data about DIC diagnosis based on any of the three criteria (ISTH-DIC, JAAM-DIC, and SIC); 4) studies that reported the relationship between DIC diagnosis and at least one of the following outcomes: sensitivity or specificity or AUC to predict mortality. Studies were excluded if they were conference abstracts, case studies, reports of the same study, reviews, studies which did not report sensitivity, specificity, or data to calculate the score performance characteristics.

## Study selection

Titles and abstracts of records identified from databases were independently screened by GT and ET using online Covidence software (Covidence systematic review software, Veritas Health Innovation, Melbourne, Australia). Available at www.covidence.org.). Then, the full texts of each potentially eligible article were read to identify the final list of studies for analysis. Any disagreement was resolved through discussion.

## Data extraction and quality assessment

After creating common data extraction sheet on Covidence, two reviewers (GT and ET) extracted data independently. The following data were extracted from the original studies: first author; year of publication; country of origin; study design; department; age; mortality rate; objective of the study; site of infection; score evaluated. When there was missing information, we contacted the respective corresponding authors. The primary outcome was mortality (in hospital or 28/30 days mortality). The secondary outcome was diagnosis of DIC.

A PROBAST (Prediction model Risk Of Bias ASsessment Tool) was used to assess the risk of bias of the included studies [24]. Consensus on the risk of bias was sought by two reviewers (GT and ET). A detailed quality assessment is provided in S2 Table.

## Data analysis

Data were extracted in Microsoft Excel format, followed by analysis using STATA version 17·0 statistical software (STATA Corp LLC, Texas, USA), available at Stata | StataCorp LLC, and Open Meta [Analyst]- CEBM @ Brown [25]. A forest plot was used to see the pooled effect size and effect of each study with their confidence interval (CI) to provide a visual summary of score positivity, mortality rate among positive cases, and sensitivity and specificity of the scores to predict mortality. A random-effect meta-analysis was performed with a 95% confidence interval (CI). A summary table was used to describe the characteristics of the included studies. Subgroup analysis was conducted based on geographic region, and stages of sepsis (sepsis, severe sepsis, and septic shock). Statistical significance was considered at $P < 0·05$.

# Results

## Search results

In total, 5557 studies were initially identified from four databases (1882 studies from Embase, 1753 studies from Scopus, 1080 studies from Web of science, and 842 studies from Medline). After removing 2553 duplicates, 3004 studies were eligible for screening. Out of these studies, 2914 were excluded based on title and abstract screening. A total of 90 studies underwent full-text review, out of which 69 studies were excluded for the reasons indicated in Fig 1. Finally, a total of 21 studies were eligible for the systematic review and meta-analysis.

## Characteristics of included studies

As presented in Table 1, all studies were published between 2009 and 2024. The number of patients per study ranged from 79 to 1895 and the overall mortality rate in each study ranged from 7.7% to 62.5%. Nine studies were conducted in Japan, and five studies were conducted in China and four were in South Korea, while the remaining were conducted in France (n = 1), Italy (n = 1), and Germany (n = 1). Of the 21 studies, 15 studies were retrospective and the remaining six were prospective by design. The quality assessment demonstrated that most of the studies were with low risk of bias except three studies that are reported to have high risk of bias. The detailed PROBAST assessment is presented in S2 Table. Majority of the studies were

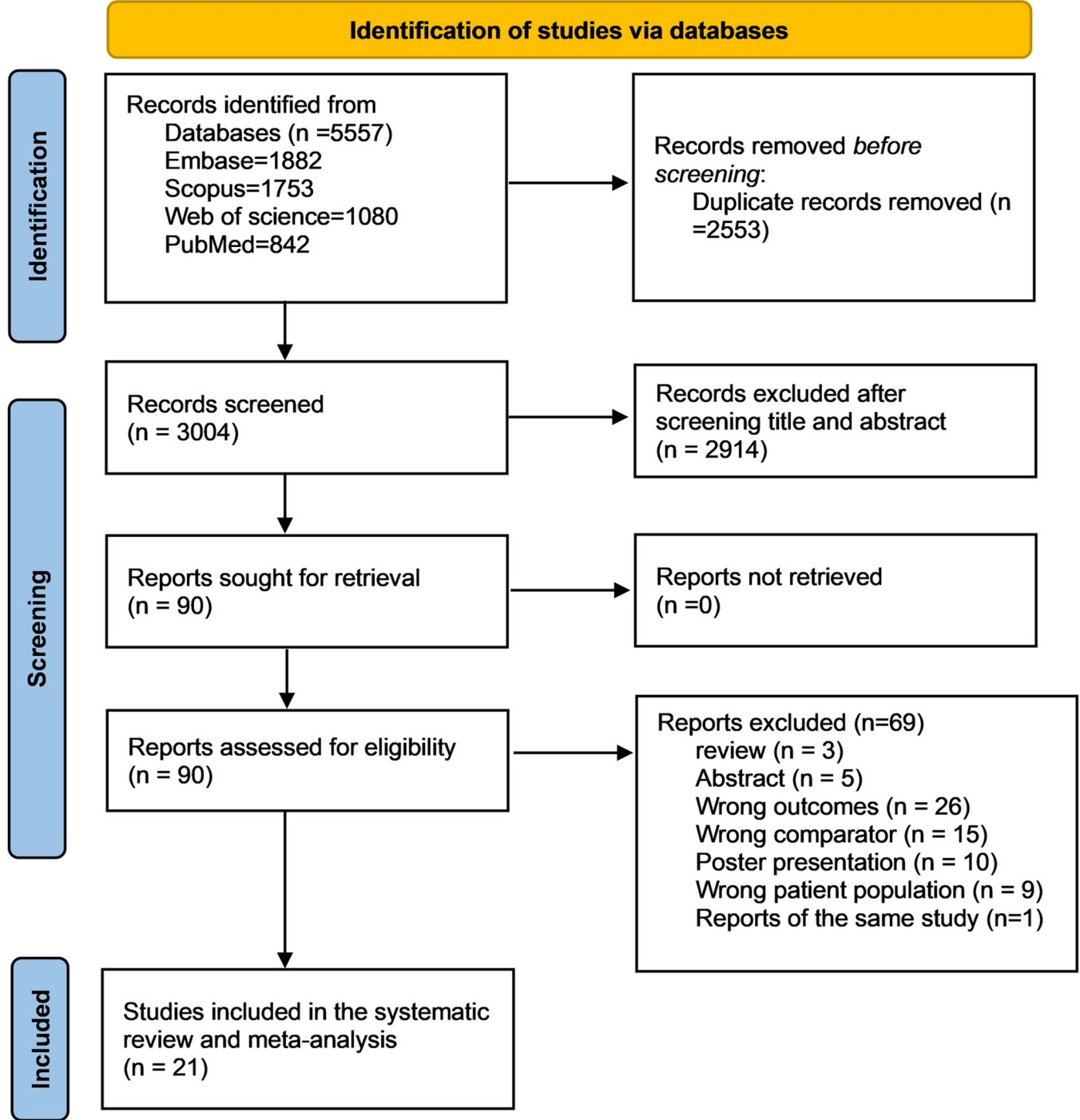

**Fig 1. PRISMA flowchart for the systematic review and meta-analysis detailing the database searches, the number of abstracts screened, and the full texts retrieved.**

conducted in the ICU setting and the mean/median age of the patients ranged from 1 to 80 years. The two most frequently reported site of infection across studies were respiratory tract infection and intra-abdominal infection.

**Table 1. Characteristics of the included studies.**

| Author and year of publication (reference) | Country | Type of study | Department | Age in Mean ± SD or Median (interquartile range) | Number of death/ Sample size (%) | Objective of the studies | Site of infection | Scores evaluated |
|---|---|---|---|---|---|---|---|---|
| Chen 2023 [26] | China | Retrospective | General wards and ICU | 65 (48–76) | 132/452 (29.2) | To evaluate prognostic performance of SIC, JAAM, and ISTH overt DIC criteria | Intra-abdominal = 230 Respiratory = 112 Bone and soft tissue = 31 Bloodstream = 26 Urinary tract = 22 Others = 52 | SIC, JAAM, and ISTH |
| Ding 2018 [18] | China | Retrospective | ICU | 65 (53.3–76) | 109/252 (43.3) | To evaluate diagnostic and prognostic performance of SIC | Abdominal = 174 Respiratory = 27 Urinary tract = 11 Enterogenic = 8 Esophageal rapture = 6 Others = 25 | ISTH and SIC |
| Gando 2009 [15] | Japan | Retrospective | ICU | 66.2± 14.3 (ISTH-) 61.4±17 (ISTH+) | 52/166 (31.3) | To test if JAAM DIC constitutes a dependent continuum to the ISTH overt DIC | Not reported | ISTH and JAAM |
| Gando 2013 [20] | Japan | Prospective | ICU | 69±16.5 | 184/624 (29.5) | To validate the JAAM DIC criteria | Not reported | ISTH and JAAM |
| Ha 2016 [27] | South Korea | Retrospective | ICU | Survivors = 67.0 (54.0–75.0) Non-survivors = 67.0 (50.0–72.0) | 31/100 | To compare the performance of DIC diagnostic criteria | Respiratory = 49 Gastrointestinal = 21 MS & skin = 8 Urinary tract = 6 CNS = 1 Heart = 1 Unkown = 14 | ISTH and JAAM |
| Helms 2020 [28] | France | Prospective | ICU | 69 (60–77) | 207/582 (35.6) | To assess the performances of the DIC scoring systems | Respiratory = 236 Urinary tract = 102 Abdominal = 89 Other = 75 | ISTH, JAAM, and SIC |
| Iba 2017 [16] | Japan | Retrospective | Emergency and ICU | Survivors = 70 (58–78) non-survivors = 73 (62–80) | 504/1498 (33.6) | To compare the SIC and JAAM-DIC criteria | Not reported | JAAM and SIC |
| Iba 2018 [21] | Japan | Retrospective | Emergency and ICU | Survivors = 74 (67–82) non-survivors = 80 (68–87) | 93/409 (22.7) | To compare the prevalence and mortality of patients identified using the SIC or JAAM-DIC criteria. | Not reported | JAAM and SIC |
| Iba 2020 [19] | Japan | Retrospective | ICU | Survivors = 73(65–80) Non-survivors = 78.5 (68–86) | 76/332 (22.9) | To validate the SIC diagnostic criteria and examine the relationship between SIC and ISTH overt DIC | Not reported | ISTH and SIC |
| Jhang 2018 [29] | South Korea | Retrospective | ICU | 9.8 (0–24) | 13/89 (14.6) | To evaluated the outcome predictability of DIC scores in critically ill children with septic shock | Not reported | ISTH and JAAM |
| Kim 2022 [30] | South Korea | Retrospective | Emergency | 65.8–12.5 | 52/295 (17.6) | To investigate the risk factors as early predictors for DIC development after admission in septic shock patients with non-overt DIC | Respiratory = 94 Genitourinary = 54 Gastrointestinal = 32 Hepatobiliary = 86 | ISTH and JAAM |

*(Continued)*

**Table 1.** (Continued)

| Author and year of publication (reference) | Country | Type of study | Department | Age in Mean ± SD or Median (interquartile range) | Number of death/ Sample size (%) | Objective of the studies | Site of infection | Scores evaluated |
|---|---|---|---|---|---|---|---|---|
| Masuda 2020 [31] | Japan | Prospective | Emergency and ICU | NO ISTH DIC = 70.2 (42.0–91.0) ISTH DIC = 75.2 (62.0–91.0) | 37/107 (34.6) | To propose a simple set of DIC criteria with coagulation-fibrinolysis markers | Not reported | ISTH and JAAM |
| Ogura 2014 [32] | Japan | Prospective | ICU | 69 ± 17 | 144/624 (23.1) | to evaluate the prognostic factors of severe sepsis in Japan | Respiratory = 261 Abdominal = 133 Urinary = 78 Skin/Soft tissue = 78 Meningitis = 15 Catheter-related = 11 Bone/Joint = 10 Wound = 10 Infective endocarditis = 3 Other = 25 | JAAM |
| Oh 2010 [33] | South Korea | Prospective | ICU | 63 (20–93) | 35/135 (25.9) | To analyzed the validity of modified non-overt DIC criteria | Respiratory = 54 Hepatobiliary = 29 Urinary tract = 23 Others = 22 | ISTH |
| Schmoch 2023 [22] | German | Retrospective | ICU | SIC positive = 69 [55–75] SIC negative = 68 [55–75] | 20/259 (7.7) | To validate the performance of SIC score | Not reported | SIC |
| Tullo 2024 [34] | Italy | Retrospective | Emergency | 79 (69–84) | 82/357 (23.0) | To assess the predictive role of SIC | Urinary tract = 174 Respiratory = 49 Abdomen = 31 Blood stream = 11 Unknown = 55 Other = 37 | SIC |
| Umemura 2016 [35] | Japan | Prospective | Emergency and ICU | 72 (63–77) | 13/79 (16.5) | Evaluate the predictive value of molecular markers in sepsis-induced DIC | Abdomen = 25 Urinary tract = 23 Soft tissue = 15 Respiratory = 11 Others = 5 | ISTH and JAAM |
| Wang 2022 [36] | China | Retrospective | ICU | 55(42–67) | 90/296 (30.4) | To validate diagnostic performance of two criteria | Respiratory = 190 Abdomen = 68 Urinary tract = 36 Others = 38 | ISTH and SIC |
| Xiang 2021 [37] | China | Retrospective | ICU | 1 (0.25, 2.3) | 40/91(43.9) | To explore the clinical value of pSIC score in diagnosis and prognosis of SIC in children. | Not reported | ISTH and SIC |
| Yamakawa 2019 [38] | Japan | Retrospective | ICU | 72 (62–80) | Not reported/ 1892 | To evaluates the significance of SIC and SAC criteria compared with the ISTH overt (DIC) and JAAM DIC criteria | Abdomen = 631 Respiratory = 471 Urinary tract = 310 Bone/soft tissue = 241 CNS = 49 Others/unknown = 190 | ISTH, JAAM, and SIC |
| Yin 2014 [39] | China | Prospective | Emergency | 72 (60–78) | 225/680 (33.1) | To investigate the value of the ISTH score in the evaluation of the 30-day mortality | Respiratory = 474 Abdomen = 156 Urinary tract = 24 CNS = 22 Skin/soft tissue = 4 | ISTH |

Sixteen studies reported the performance of ISTH-DIC score involving 4466 patients; thirteen studies (3862 patients) reported the performance of JAAM-DIC score, and ten studies (4714 patients) reported the performance of SIC score. Seven studies compared the performance of ISTH and JAAM score, four studies compared ISTH and SIC, two studies compared JAAM and SIC, and three studies compared all the three scores. The SOFA score of the included studies ranged from 2 to 13 in survivors and 4 to 16 in non-survivors. The Acute Physiology and Chronic Health Evaluation II (APACHE II) score ranged from 9 to 27 in survivors and 12 to 33 in non-survivors. The detailed clinical and laboratory values of the included studies are presented in S3 Table.

## Score positivity and mortality rate

The pooled proportion of cases diagnosed as positive using ISTH-DIC criteria was 28% (95% CI: 23–33%). The pooled proportion of positive cases was higher when using the JAAM-DIC and SIC criteria: 55% (95% CI:42–68%), and 57% (95% CI: 42–72%), respectively (Fig 2). The pooled mortality rate among patients diagnosed as positive using ISTH-DIC, JAAM-DIC, and

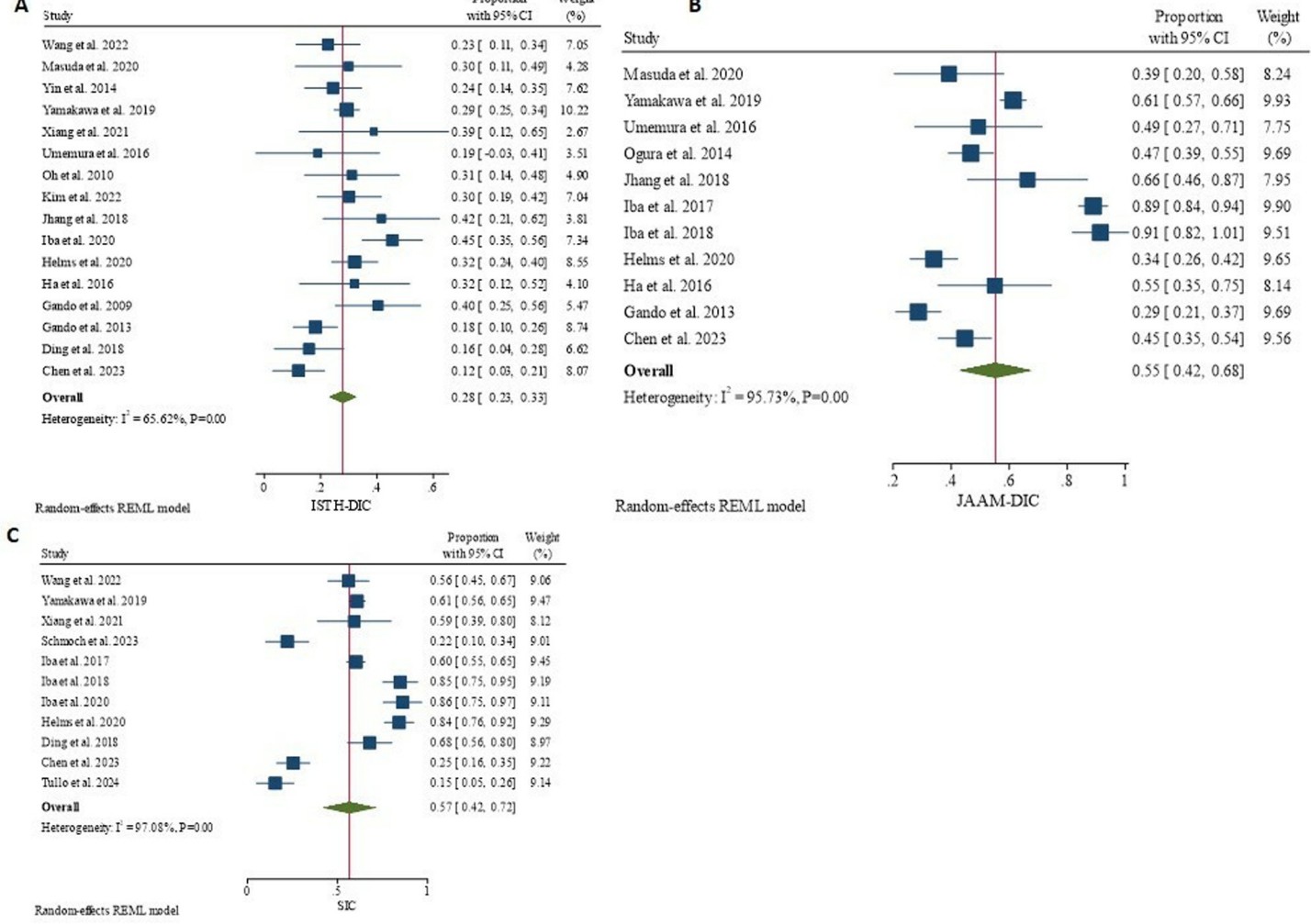

**Fig 2.** Proportion of ISTH DIC positive (A), JAAM DIC positive (B), and SIC positive (C) cases in sepsis.

SIC were 44% (95% CI:34–53%), 37% (95% CI: 29–45%), and 35% (95% CI: 30–41%), respectively (Fig 3). Three studies [26, 28, 38] reported the proportion of positive scores and mortality rate for the three scores simultaneously.

For the three studies that analyzed proportion of score positivity and mortality rates of all the three scores on the same population, a separate meta-analysis was undertaken to see if the effect sizes are different from the previous meta-analysis that involved all studies. Comparable results were obtained for ISTH-DIC and JAAM-DIC scores. A pooled proportion of positive scores and mortality rates was 25% (95% CI: 13–37%) and 49% (95% CI: 34–64%), respectively for ISTH-DIC. It was 47% (95% CI: 31–63%) and 37% (95% CI: 28–46%), respectively for JAAM-DIC. For SIC, similar values were obtained; 57% (95% CI: 24–90%) and 35% (95% CI: 30–39%), respectively (Fig 4).

A pooled proportion of patients diagnosed as both SIC and ISTH positive was 41% (95% CI: 33–49%). The proportion of patients diagnosed as both JAAM and ISTH positive was 49% (95% CI: 37–62%) (Fig 5).

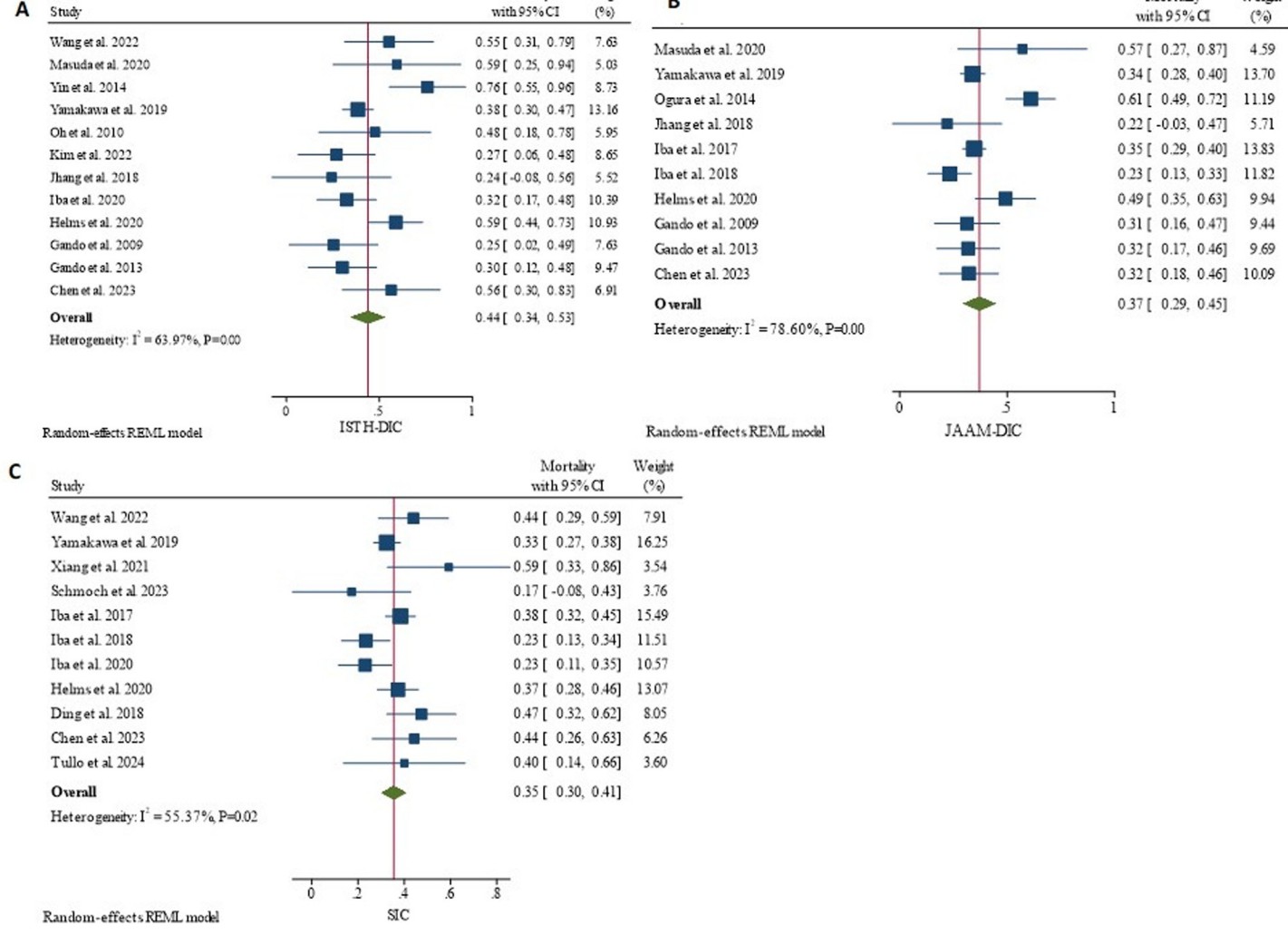

**Fig 3.** Mortality rate among ISTH DIC positive (A), JAAM DIC positive (B), and SIC positive (C) individuals.

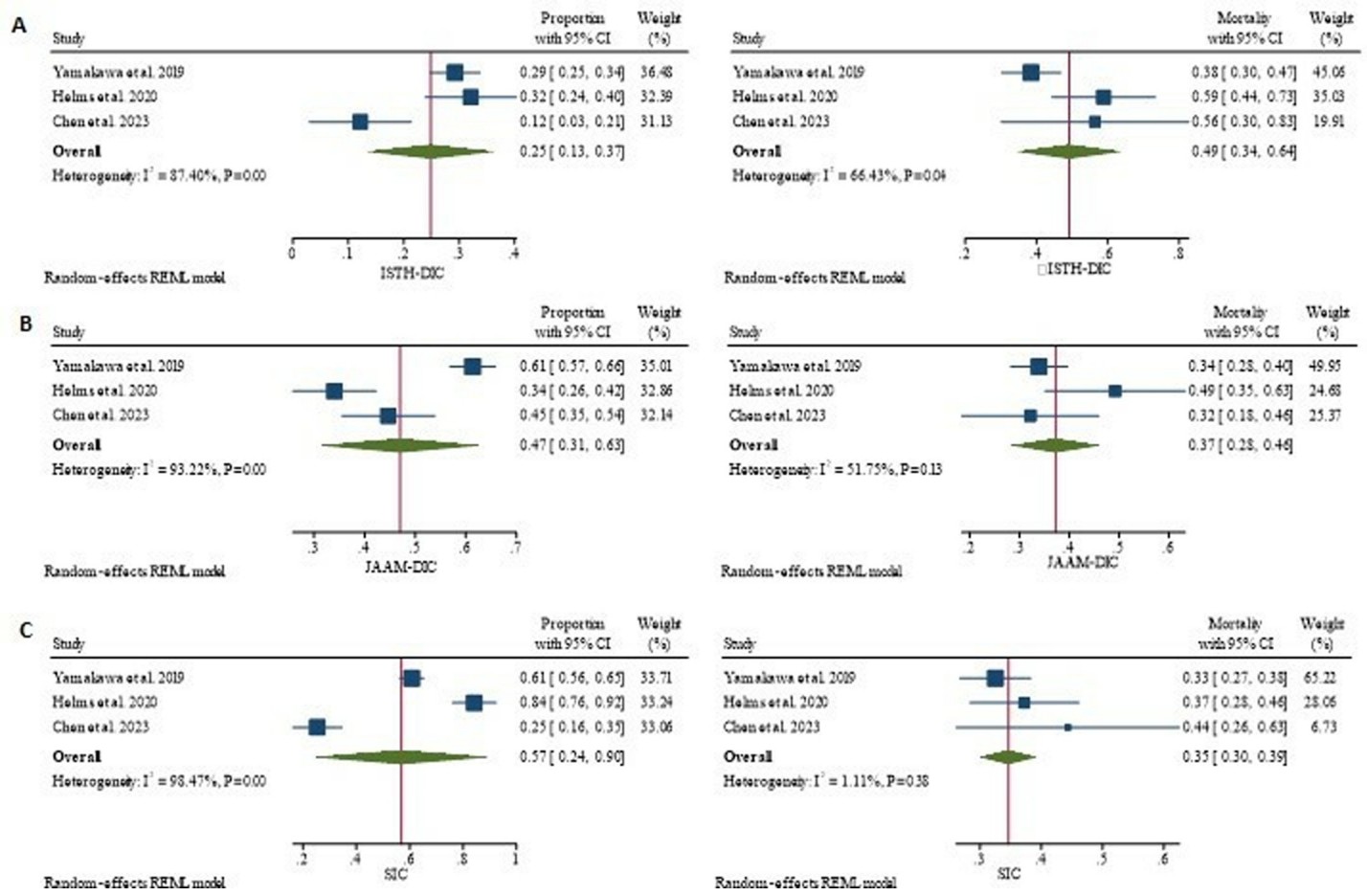

**Fig 4.** Proportion of positive score and mortality rates of ISTH-DIC (**A**), JAAM-DIC (**B**), and SIC (**C**), across the same studies.

A separate meta-analysis was undertaken to observe subgroup effect of geographical region and stages of sepsis. The pooled proportion of positive cases using ISTH-DIC was 22% (95% CI: 16–29%) in China, 32% (95% CI: 15–84%) in Europe, 31% (95% CI: 21–40%) in Japan, and 32% (95% CI: 24–40%) in South Korea (S1 Fig). The proportion of ISTH-DIC positive was 30% (95% CI: 22–38%), 25% (95% CI: 17–32%), and 32% (95% CI: 26–39%) in sepsis, severe sepsis, and septic shock patients, respectively. Similar sub-group analysis could not be conducted for the other criteria due to limited availability of data.

## Mortality prediction

As a predictor of mortality, the pooled sensitivity of ISTH-DIC across all included studies was 0.43 (95% CI: 0.34–0.52); the pooled specificity was 0.81 (95% CI: 0.74–0.87) (Fig 6A). The pooled sensitivity and specificity of JAAM-DIC for predicting mortality in sepsis patients were 0.73 (95% CI: 0.57–0.85) and 0.46 (95% CI: 0.28–0.65), respectively (Fig 6B). SIC pooled sensitivity and specificity were 0.71 (95% CI: 0.57–0.82) and 0.49 (95% CI: 0.31–0.66), respectively (Fig 6C). Two studies [28, 38] reported the sensitivity and specificity of the three scores simultaneously. In the subgroup analysis, better sensitivity of JAAM-DIC was observed in sepsis patients at 0.87 (95% CI: 0.67–0.96) compared to severe sepsis and septic shock patients, but the specificity was lower at 0.26 (95% CI: 0.06–0.62) (S2 Fig).

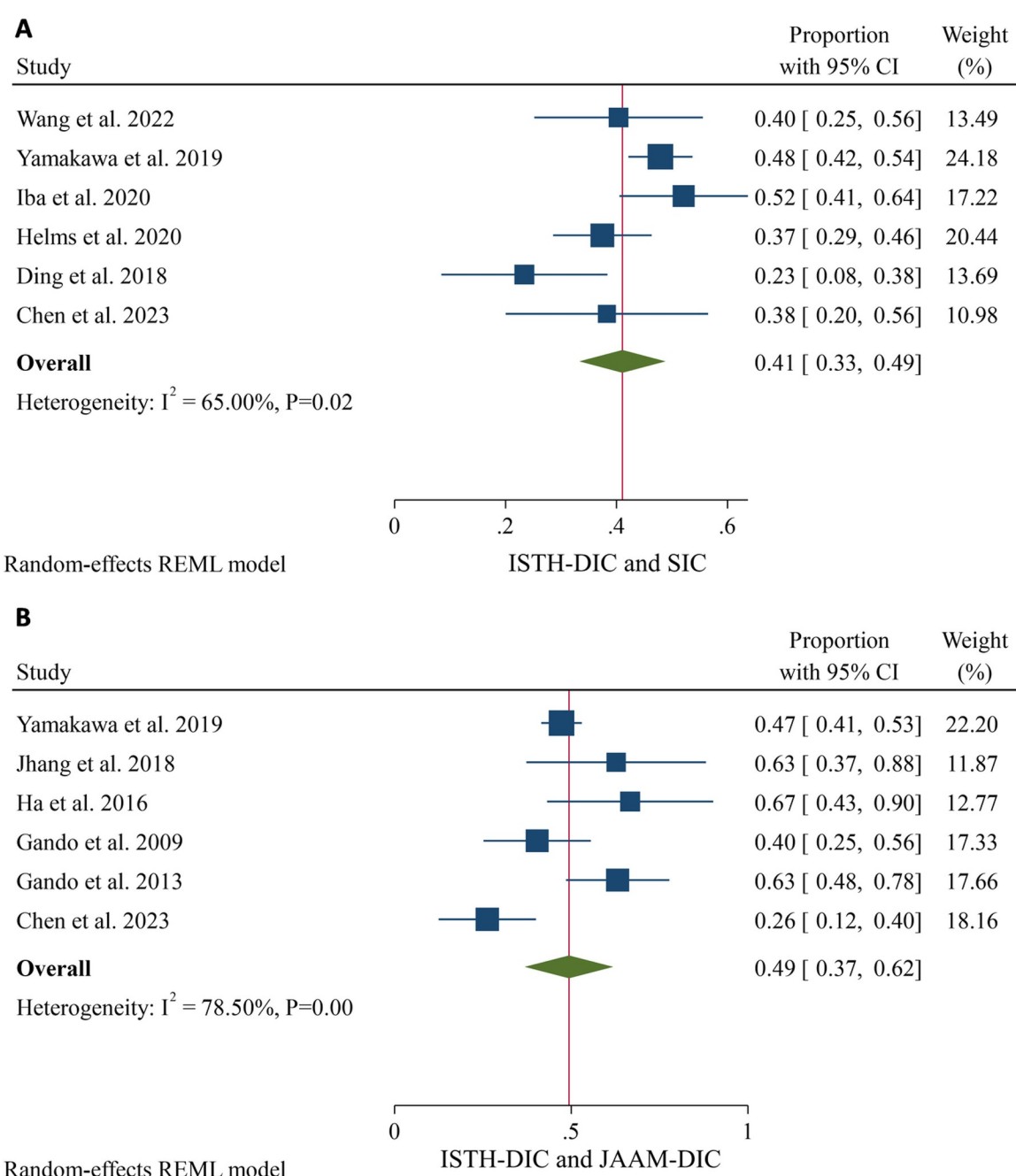

**Fig 5.** The proportion of patients diagnosed as SIC and ISTH positive (A), and patients diagnosed as JAAM and ISTH positive (B).

## Discussion

Disseminated intravascular coagulation is a common complication in sepsis patients which exacerbates clinical outcomes. The magnitude of DIC in sepsis highly depends on the severity of the disease and diagnostic approaches utilized. In this systematic review and meta-analysis, the proportion of DIC and coagulopathy based on the ISTH-DIC, JAAM-DIC, and SIC criteria and the performance of these criteria to predict mortality in sepsis patients were reported. A higher pooled proportion of positive scores was reported when using the JAAM-DIC and SIC

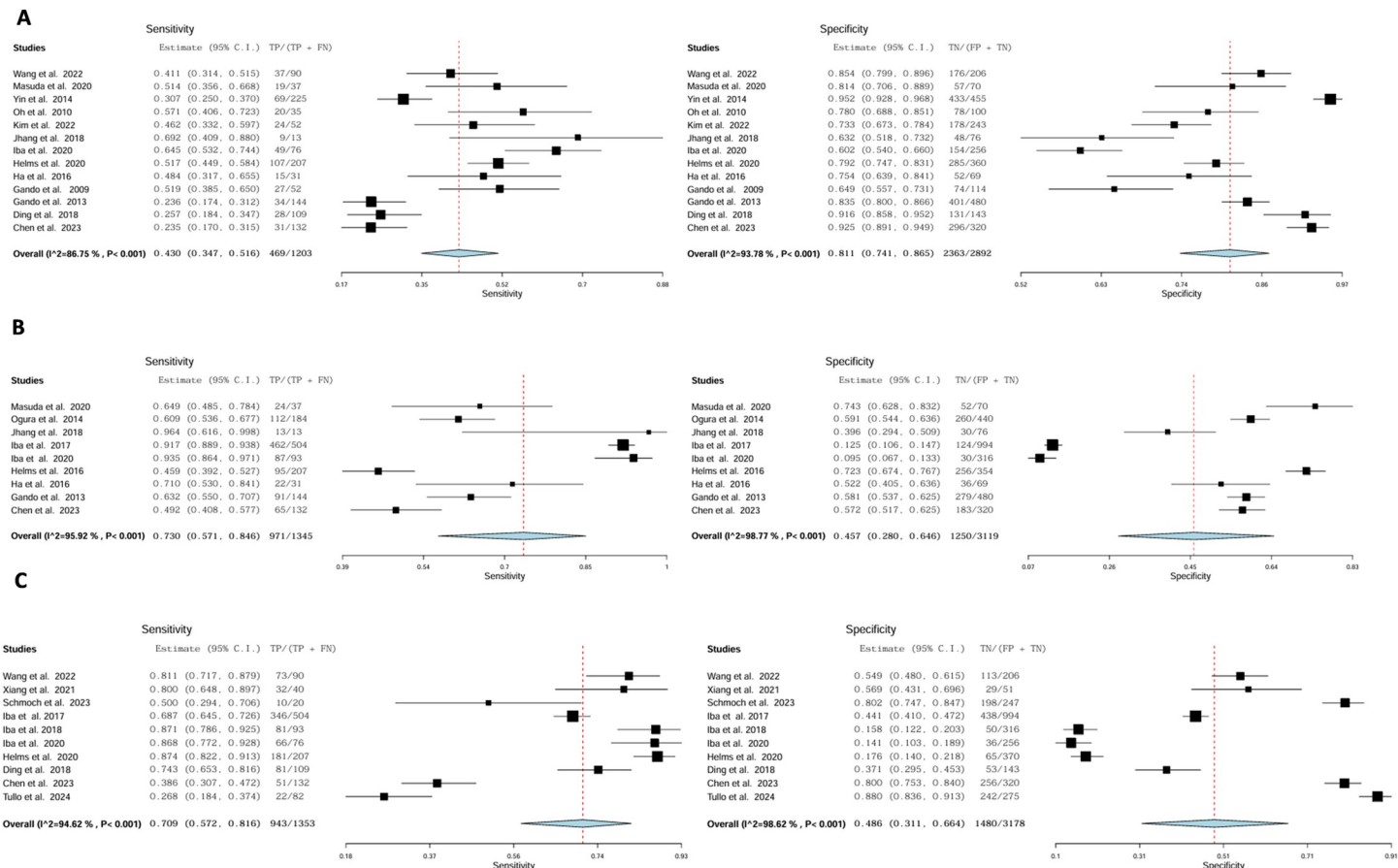

**Fig 6.** Sensitivity and specificity to predict 28 day mortality A: ISTH DIC, B: JAAM DIC, C: SIC.

criteria compared to the ISTH-DIC criteria. On the contrary, the pooled mortality rate was higher in ISTH-DIC-positive patients compared to patients with positive scores of JAAM-DIC and SIC. This finding was comparable when we conducted a separate meta-analysis only on the three studies that reported positive scores and mortality rates of all the three scores on the same population.

Though there is no gold standard method to diagnose DIC, overt-DIC criteria by ISTH has been recognized as a global standard [2]. These criteria best identify patients who are at an advanced (possibly irreversible) stage of coagulopathy [37] and hence have relatively lower sensitivity. The present study reported that the proportion of sepsis patients with positive ISTH-DIC score accounted for 28%. This is lower than the pooled proportion obtained by using JAAM-DIC and SIC criteria. Many of the overt-DIC positive patients are thought to be at an advanced stage of coagulopathy and might not benefit from anticoagulant therapy [40]. Congruent to this is the higher mortality rate observed in patients with positive ISTH-DIC score compared to the other scores. Overt DIC is often related to severe illness and increased risk of death in sepsis patients worsening patient outcomes [41].

The JAAM-DIC criteria were reported to diagnose DIC at an earlier stage in sepsis patients and were able to diagnose all patients with ISTH-DIC even earlier at sepsis diagnosis [20]. In our study, the pooled proportion of septic patients with positive JAAM-DIC score was higher at 55%. Moreover, the proportion of JAAM-DIC positive patients who are also ISTH-DIC positive

was 49%. A study that involved 1895 sepsis patients reported that those with positive JAAM-DIC were twice higher in number compared to those with ISTH-DIC [6]. Earlier diagnosis of patients with DIC in sepsis is important to identify patients who can benefit from anticoagulant therapy. According to a multicenter validation study in sepsis patients, the JAAM-DIC criteria demonstrated good prognostic value in predicting multiorgan dysfunction syndrome and poor patient outcomes identifying more patients who require treatment [20].

The SIC criteria were particularly proposed for patients with sepsis by incorporating the latest definition of sepsis as a SOFA score of 2 or greater [16]. Compared to the other two DIC criteria, SIC exhibited higher sensitivity. More than half of sepsis patients had positive SIC scores in the current meta-analysis which is two times greater compared to the ISTH-DIC criteria. Moreover, the pooled proportion of SIC positive patients who also are ISTH positive was 41% and almost all the ISTH-DIC positive patients were SIC positive. Previous studies also reported that all the ISTH-DIC positive patients were diagnosed as SIC positive, and positive SIC score came ahead before positive ISTH-DIC score in every case [42]. In the present study, sufficient data could not be obtained to evaluate the performance of JAAM_DIC and SIC to predict the occurrence of ISTH-DIC. The SIC criteria was primarily designed to predict the occurrence of ISTH DIC and should be used as a screening tool. In contrast, ISTH DIC is established as the definitive diagnosis by excluding other conditions that mimic DIC. Thus, a two-step integrated scoring algorithm has been proposed by ISTH, which encompasses early screening of patients using SIC and subsequent calculation of the ISTH-DIC score for those who met the SIC criteria at the first step [43].

According to the subgroup analysis, the proportion of positive ISTH-DIC score was lower in the studies from China. This may be related to differences in disease severity among study participants. As shown in the S2 Table, the SOFA scores of studies from China were lower, ranging from 4 to 8.5, compared to the scores from other countries, which ranged from 5 to 13.6. The proportion of ISTH-DIC positive score among severe sepsis patients was found to be lower compared to patients with sepsis and septic shock. This might be explained by the sepsis definition used in the studies. Severe sepsis is defined according to Sepsis-2 definition, which captures relatively mild infections and non-infectious conditions though it has high sensitivity [44]. Sepsis and severe sepsis are sometimes used interchangeably to indicate the presence of infection complicated by organ dysfunction [45].

DIC is associated with the severity of sepsis and is involved in its pathogenesis [46]. Though there are dedicated clinical scores that predict disease severity and outcomes of the patient [47], evaluation of DIC scores for their performance in predicting patient outcomes would be valuable. In the present study, the pooled sensitivity of the ISTH-DIC score to predict mortality in sepsis patients is lower compared to the other scores. However, the specificity was higher than the other scores. The JAAM-DIC and SIC scores exhibited higher sensitivity and lower specificity in predicting mortality in sepsis patients compared to the ISTH-DIC criteria. Similar trends were observed in two studies that compared the three scoring systems simultaneously.

The sub-group analysis showed increased sensitivity of the JAAM-DIC score to predict mortality in sepsis patients compared to severe sepsis and septic shock. These findings are generally related to the proportion of corresponding positive scores in sepsis patients. A higher proportion of positive scores is largely related to higher sensitivity and lower specificity [48].

The strength of this study is that it provided a comprehensive review of existing literatures. Understandably, there is a lack of meta-analysis that compared different DIC scores due to the absence of a gold standard method and lack of uniformity across regions to diagnose DIC. This makes the present study the first one to compare different DIC scoring systems through meta-analysis. However, our review has several limitations. First, there was significant heterogeneity between the included studies. Second, there was a lack of uniformity concerning the

diagnostic criteria for sepsis across the included studies. Moreover, studies were not equitably covered across regions of the world, whereby many of the studies were from Asia, and there were limited studies from other regions. The study was not patient-level meta-analysis that limited conducting additional useful analysis.

In conclusion, the three DIC and coagulopathy scores at hand yielded varying score positivity and mortality prediction performance in sepsis patients. The SIC and JAAM-DIC scores exhibited higher positivity rate and predict patient outcomes, and thus are valuable to identify patients for possible treatment at an early stage. The ISTH-DIC score perhaps identified patients at later stages and demonstrated better specificity in predicting disease outcomes. Thus, early identification of patients using the SIC and JAAM-DIC scores and later confirmation using the ISTH-DIC score would be beneficial approaches for improved management of patients with sepsis.

## Supporting information

**S1 Checklist. PRISMA 2020 checklist.**
(DOCX)

**S1 File. List of excluded studies at full text read stage.**
(DOCX)

**S1 Dataset. Extracted data from included studies and used for all analyses.**
(CSV)

**S1 Table. Database search results.**
(DOCX)

**S2 Table. Assessment of risk of bias using Prediction model Risk Of Bias Assessment Tool (PROBAST).**
(DOCX)

**S3 Table. Clinical and laboratory values of the included studies.**
(DOCX)

**S1 Fig. Subgroup analysis of score positivity of ISTH-DIC by geographical region and sepsis stages.**
(TIF)

**S2 Fig. Subgroup analysis of JAAM-DIC sensitivity and specificity to predict mortality, by sepsis stages.**
(TIF)

## Author Contributions

**Conceptualization:** Girum Tesfaye Kiya.

**Data curation:** Girum Tesfaye Kiya, Edosa Tadasa.

**Formal analysis:** Girum Tesfaye Kiya, Gemeda Abebe, Edosa Tadasa, Elsah Tegene Asefa.

**Funding acquisition:** Girum Tesfaye Kiya, Edosa Tadasa, Elsah Tegene Asefa.

**Investigation:** Girum Tesfaye Kiya, Edosa Tadasa, Gedion Milkias, Elsah Tegene Asefa.

**Methodology:** Girum Tesfaye Kiya, Gemeda Abebe, Zeleke Mekonnen, Edosa Tadasa, Gedion Milkias, Elsah Tegene Asefa.

**Project administration:** Girum Tesfaye Kiya, Gemeda Abebe, Zeleke Mekonnen.

**Software:** Girum Tesfaye Kiya.

**Supervision:** Girum Tesfaye Kiya, Gemeda Abebe, Zeleke Mekonnen, Elsah Tegene Asefa.

**Validation:** Girum Tesfaye Kiya, Gemeda Abebe, Zeleke Mekonnen, Edosa Tadasa, Gedion Milkias.

**Visualization:** Girum Tesfaye Kiya.

**Writing – original draft:** Girum Tesfaye Kiya.

**Writing – review & editing:** Gemeda Abebe, Zeleke Mekonnen, Edosa Tadasa, Gedion Milkias, Elsah Tegene Asefa.

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
