## [Decision Letter · Decision Letter 0]

14 Oct 2024

PONE-D-24-39032A comparison of Disseminated Intravascular Coagulation Scoring Systems and Their Performance to Predict Mortality in Sepsis patients: A Systematic Review and Meta-analysisPLOS ONE

Dear Dr. Kiya,

Thank you for submitting your manuscript to PLOS ONE. After careful consideration, we feel that it has merit but does not fully meet PLOS ONE’s publication criteria as it currently stands. Therefore, we invite you to submit a revised version of the manuscript that addresses the points raised during the review process.

We look forward to receiving your revised manuscript.

Kind regards,

Mehmet Baysal

Academic Editor

PLOS ONE

Journal Requirements:

“GTK, ET, GM, and ETA are supported by the Research and Innovation Director (RID) office with Mega Research Fund Scheme (2022-2025). The researchers were independent of their sources of support, which had no role in this study.”

5. As required by our policy on Data Availability, please ensure your manuscript or supplementary information includes the following:

Reviewers' comments:

Reviewer's Responses to Questions

**Comments to the Author**

1. Is the manuscript technically sound, and do the data support the conclusions?

Reviewer #1: Yes

Reviewer #2: Yes

Reviewer #3: Partly

Reviewer #4: Yes

2. Has the statistical analysis been performed appropriately and rigorously? 

Reviewer #1: Yes

Reviewer #2: Yes

Reviewer #3: No

Reviewer #4: I Don't Know

3. Have the authors made all data underlying the findings in their manuscript fully available?

Reviewer #1: Yes

Reviewer #2: Yes

Reviewer #3: Yes

Reviewer #4: Yes

4. Is the manuscript presented in an intelligible fashion and written in standard English?

Reviewer #1: Yes

Reviewer #2: Yes

Reviewer #3: Yes

Reviewer #4: Yes

5. Review Comments to the Author

Reviewer #1: The paper is a systematic review and meta-analysis that focuses on comparing three different scoring systems for disseminated intravascular coagulation (DIC) in predicting mortality among sepsis patients. The study uses four databases: Medline (via PubMed), Scopus, Embase, and Web of Science, and search terms of 'Disseminated Intravascular Coagulation,' 'coagulopathy,' and the 'ISTH-DIC,' 'JAAM-DIC,' and 'SIC' scoring systems. They then performed a random-effects meta-analysis and conducted subgroup analyses based on geographic region and sepsis stages.

This methodology identified 21 studies for a total of 9319 sepsis patients. DIC positivity rates for the different scoring systems were 28% for ISTH-DIC, 55% for JAAM-DIC, and 57% for SIC. ISTH-DIC-positive patients had a pooled mortality rate of 44%, while JAAM-DIC and SIC had pooled mortality rates of 37% and 35%. The study also evaluated the predictive power of these scoring systems, finding that ISTH-DIC had a sensitivity of 0.43 and specificity of 0.81 for predicting mortality. In comparison, JAAM-DIC showed a higher sensitivity of 0.73 but lower specificity of 0.46, and SIC had a sensitivity of 0.71 with specificity of 0.49. The authors also perform subgroup analyses based on geographic region and sepsis stage, which show variations in score positivity. This supports the notion that different criteria have varying performance depending on patient populations and sepsis severity.

The authors conclude that the JAAM-DIC and SIC scores exhibit higher sensitivity for identifying patients with coagulopathy at an earlier stage, increasing their relevance for early treatment interventions. Conversely, the ISTH-DIC score, with its higher specificity, may not be useful until later in the disease process. Thus, an integrated approach, where early identification is done using JAAM-DIC or SIC scores and later confirmation is performed with the ISTH-DIC score, is recommended for better management of sepsis patients.

The data are clearly presented, the writing is clear and grammatical. I do not have suggestions for improvement, however, the overall clinical impact of these findings is limited.

Reviewer #2: This is a systematic review and meta-analysis of observational studies that report DIC scores for patients with sepsis. It reports the pooled positivity rates for three scores and the sensitivity and specificity of each for mortality.

My specific suggestions are below:

Methods

Pg 5: "Inclusion and exclusion criteria" - please clarify ..." 3) studies that describe data about DIC diagnosis based on any of the three criteria..." [note the words "any of" added]

Results

Page 16: it is not clear what the denominator is in these analyses. Are you showing the number of patients who are positive for both scores as a proportion of all the patients who had both scores measured; or e.g. the number of JAAM/ISTH dual positive as a proportion of all the JAAM positive (who also have ISTH measured)? This should be clarified.

If the data permit, I suggest calculating the number of patients with JAAM positive and SIC positive respectively as a proportion of those who were ISTH positive (and had the other score measured); also to show the number of patients with JAAM negative and SIC negative as a proportion of those who were ISTH negative. i.e. how sensitive and specific were JAAM and SIC for picking up DIC as defined by ISTH positivity? If these data are not extractable from the studies, then this should be mentioned.

If data permit, it would be helpful to calculate the mortality rate in the group that was positive for JAAM or SIC respectively and also negative for ISTH, and compare against the mortality rate in the group that was positive for ISTH. i.e. does calculation of the JAAM or SIC score improve discrimination for mortality above ISTH, or do they simply include more patients indiscriminately?

Discussion

The comparative heterogeneity of the three scores between studies should be mentioned. The heterogeneity of the positivity rate for ISTH score seems lower than for the other scores. This is likely at least partially due to the fact that the rate is much lower.

It should be remembered, and probably mentioned in the article, that prediction of mortality is not the purpose of the scores. Clinically, the main purpose is to differentiate between DIC and other causes of coagulopathy and thrombocytopenia in patients with relatively severe illness.

Pg 21: Limitations - must include that this was not a patient-level meta-analysis. Several useful analyses could not be undertaken due to this limitation.

Pg 21: Conclusion - I do not believe this meta-analysis is able to determine sensitivity of the scores for detecting DIC/coagulopathy given a lack of gold standard. Therefore, the conclusion should simply state that the JAAM and SIC scores have higher positivity rates than ISTH. The meta-analysis does show higher sensitivity to mortality, with concomitant lower specificity. Patient-level analysis may be required to determine whether the discrimination is improved by JAAM and SIC. It is not clear that using SIC or JAAM to identify DIC patients at an early stage will be beneficial in terms of patient outcomes - at most you can state that SIC or JAAM could be used for early identification followed by confirmation using ISTH score at later stages of the clinical course.

Supplementary Figures - the Figure legends should be more specific and describe what the figure shows without need to refer back to the main text i.e. is the Figure of score positivity or mortality prediction.

Reviewer #3: The authors aimed to evaluate what DIC score system performs better in sepsis patients. The matter is relevant and interesting. The manuscript is globally well-written and easy to follow, with good standards from the methodology point of view. However, from the results sections onwards I have several concerns. Majors are:

- Figure 2 reports the pooled proportion of each positive score among the entire population. Unfortunately, only a few studies compared in the same population all the three DIC-scores. So, the proportion of each positive score is not comparable with the others because different populations were considered (figure A includes 16 studies, B 11 studies, C 11 studies (but not the same of B)). In my opinion, figure 2 does not provide any relevant information.

- Figure 3: It would have been interesting to compare the different pooled mortality rates of the same population measured with each DIC score. As above, comparing different populations does not provide relevant information on DIC-scores performance.

- There are two Figure 4. The second one provides the most relevant analysis of the manuscript. However, due to the very high heterogeneity in all sections (A-B-C) caution must be used when interpreting the results.

My suggestion is to rethink the analyses using a comparable population to evaluate the performance of each DIC-score.

Reviewer #4: Thank you for the opportunity to review that meta-analysis about different diagnostic criteria for DIC for prognostics in sepsis patients. Due to the high mortality in septic patients, it is important to have different prognostic tools to identify critically ill patients early. This is especially important in settings with limited resources as ICU beds are scarce. In my opinion the authors performed an overall well-conducted review, were only a few questions are open, before publishing it:

- According to your manuscript, you performed a search update this August. Did you include any new studies and can you add that information in the PRISMA flow-chart (how many have you found and how many did you include). Did you include only papers in English or also other languages?

- Can you state the inclusion and exclusion criteria based on the SPIDER tool?

- In the abstract you say, you performed a random-effect meta-analysis, but I did not find anything about it in the full text. Please explain, why you used this method

- Two times, you state you want to evaluate the diagnosis of DIC or identify patients with DIC. Can you change this, as you only determined the score positivity rate / assessed the proportion of DIC and coagulopathy in sepsis patients

- In the results section you wirte „Three studies …“ but only two studies are cited.

- Please write out abbreviations first time using them (also if you write them out in the abstract, you have to write it out the first time in the full manuscript as well)

- Please add the newest sepsis guideline: DOI: 10.1097/CCM.0000000000005337

- Can you add one or two sentences to the pathomechanism of DIC and are there any risk factors associated with the development of DIC

- The department of the first study is missing. If it is not available, please indicate as well.

- Please also consider including this recently published review: DOI: 10.7759/cureus.67052

- Could you please add the references for the following sentence „Though there are dedicated clinical scores that …“

- Can you compare the sensitivity and specifity to predict the mortalitiy in patients with sepsis to the dedicated clinical scores

A general recommendation for future manuscript - activate the row count to simplify reviewing the manuscript

6. PLOS authors have the option to publish the peer review history of their article (what does this mean?). If published, this will include your full peer review and any attached files.

Reviewer #1: No

Reviewer #2: **Yes: **Vinai Chander Bhagirath

Reviewer #3: No

Reviewer #4: **Yes: **Christoph Veigl

---

## [Author Response · Author response to Decision Letter 0]

7 Nov 2024

Response to reviewers

 Dear Editor and Reviewers,

We, the authors, are very thankful for your valuable and constructive comments on the manuscript. We made a point-by-point response to your comments as follows and changes were made in the manuscript using track change:

Reviewer #1: The paper is a systematic review and meta-analysis that focuses on comparing three different scoring systems for disseminated intravascular coagulation (DIC) in predicting mortality among sepsis patients. The study uses four databases: Medline (via PubMed), Scopus, Embase, and Web of Science, and search terms of 'Disseminated Intravascular Coagulation,' 'coagulopathy,' and the 'ISTH-DIC,' 'JAAM-DIC,' and 'SIC' scoring systems. They then performed a random-effects meta-analysis and conducted subgroup analyses based on geographic region and sepsis stages.

This methodology identified 21 studies for a total of 9319 sepsis patients. DIC positivity rates for the different scoring systems were 28% for ISTH-DIC, 55% for JAAM-DIC, and 57% for SIC. ISTH-DIC-positive patients had a pooled mortality rate of 44%, while JAAM-DIC and SIC had pooled mortality rates of 37% and 35%. The study also evaluated the predictive power of these scoring systems, finding that ISTH-DIC had a sensitivity of 0.43 and specificity of 0.81 for predicting mortality. In comparison, JAAM-DIC showed a higher sensitivity of 0.73 but lower specificity of 0.46, and SIC had a sensitivity of 0.71 with specificity of 0.49. The authors also perform subgroup analyses based on geographic region and sepsis stage, which show variations in score positivity. This supports the notion that different criteria have varying performance depending on patient populations and sepsis severity.

The authors conclude that the JAAM-DIC and SIC scores exhibit higher sensitivity for identifying patients with coagulopathy at an earlier stage, increasing their relevance for early treatment interventions. Conversely, the ISTH-DIC score, with its higher specificity, may not be useful until later in the disease process. Thus, an integrated approach, where early identification is done using JAAM-DIC or SIC scores and later confirmation is performed with the ISTH-DIC score, is recommended for better management of sepsis patients.

The data are clearly presented, the writing is clear and grammatical. I do not have suggestions for improvement; however, the overall clinical impact of these findings is limited.

Reviewer #2: This is a systematic review and meta-analysis of observational studies that report DIC scores for patients with sepsis. It reports the pooled positivity rates for three scores and the sensitivity and specificity of each for mortality.

My specific suggestions are below:

Methods

Pg 5: "Inclusion and exclusion criteria" - please clarify ..." 3) studies that describe data about DIC diagnosis based on any of the three criteria..." [note the words "any of" added]

Response: Thank you for the valuable comment. We added the phrase accordingly

Results

Page 16: it is not clear what the denominator is in these analyses. Are you showing the number of patients who are positive for both scores as a proportion of all the patients who had both scores measured; or e.g. the number of JAAM/ISTH dual positive as a proportion of all the JAAM positive (who also have ISTH measured)? This should be clarified.

Response: Thank you for the comment. As it is described in the characteristics of the included studies, some studies measured both scores, and few even measured all the three scores for each patient in the study. Thus, the denominator in these analyses is all patients in the respective studies.

If the data permit, I suggest calculating the number of patients with JAAM positive and SIC positive respectively as a proportion of those who were ISTH positive (and had the other score measured); also to show the number of patients with JAAM negative and SIC negative as a proportion of those who were ISTH negative. i.e. how sensitive and specific were JAAM and SIC for picking up DIC as defined by ISTH positivity? If these data are not extractable from the studies, then this should be mentioned.

Response: Thank you for the comments. Figure 4 somehow showed the proportion of positive scores both by JAAM and ISTH, as well as by SIC and ISTH. These scores are measured at the same time point and the mentioned data are not extractable for the required analysis. statement has been added in the 4th paragraph of discussion to clarify this issue as suggested.

If data permit, it would be helpful to calculate the mortality rate in the group that was positive for JAAM or SIC respectively and also negative for ISTH, and compare against the mortality rate in the group that was positive for ISTH. i.e. does calculation of the JAAM or SIC score improve discrimination for mortality above ISTH, or do they simply include more patients indiscriminately?

Response: Thank you for the valuable comments. As it has been shown, almost all ISTH-DIC positive patients are also positive for SIC and JAAM. Thus, the sole effect of SIC or JAAM positivity on mortality could not be evaluated.

Discussion

The comparative heterogeneity of the three scores between studies should be mentioned. The heterogeneity of the positivity rate for ISTH score seems lower than for the other scores. This is likely at least partially due to the fact that the rate is much lower.

Response: Thank you for the comments. Although the extent of heterogeneity is different across scores, all are with substantial heterogeneity, which is also indicated as a limitation of our study.

It should be remembered, and probably mentioned in the article, that prediction of mortality is not the purpose of the scores. Clinically, the main purpose is to differentiate between DIC and other causes of coagulopathy and thrombocytopenia in patients with relatively severe illness.

Response: Thank you for the suggestion. This notion is described in the sixth paragraph (2nd sentence) of discussion.

Pg 21: Limitations - must include that this was not a patient-level meta-analysis. Several useful analyses could not be undertaken due to this limitation.

Response: Thank you for the valuable comment. We added the suggested statement in the limitation part.

Pg 21: Conclusion - I do not believe this meta-analysis is able to determine sensitivity of the scores for detecting DIC/coagulopathy given a lack of gold standard. Therefore, the conclusion should simply state that the JAAM and SIC scores have higher positivity rates than ISTH. The meta-analysis does show higher sensitivity to mortality, with concomitant lower specificity. Patient-level analysis may be required to determine whether the discrimination is improved by JAAM and SIC. It is not clear that using SIC or JAAM to identify DIC patients at an early stage will be beneficial in terms of patient outcomes - at most you can state that SIC or JAAM could be used for early identification followed by confirmation using ISTH score at later stages of the clinical course.

Response: Thank you for the insightful comment. We agree with the comment and made correction on the same paragraph

Supplementary Figures - the Figure legends should be more specific and describe what the figure shows without need to refer back to the main text i.e. is the Figure of score positivity or mortality prediction.

Response: Thank you for the comments. We corrected figure legends based on the suggestion provided.

Reviewer #3: The authors aimed to evaluate what DIC score system performs better in sepsis patients. The matter is relevant and interesting. The manuscript is globally well-written and easy to follow, with good standards from the methodology point of view. However, from the results sections onwards I have several concerns. Majors are:

- Figure 2 reports the pooled proportion of each positive score among the entire population. Unfortunately, only a few studies compared in the same population all the three DIC-scores. So, the proportion of each positive score is not comparable with the others because different populations were considered (figure A includes 16 studies, B 11 studies, C 11 studies (but not the same of B)). In my opinion, figure 2 does not provide any relevant information.

Response: Thank you for the valuable comments. Though these population are different, all of them were sepsis patients admitted to emergency and ICU. Variables that could cause heterogeneity were also dealt in the sub-group analysis. In addition, after this comment, pooled proportion of positive score of the three studies that measured all the three DIC-scores was analyzed with separate meta-analysis. This also yielded comparable results as it is shown in figure 4 (new). SIC positivity and mortality rate are similar when comparing the values obtained from similar population with the value from different population.

- Figure 3: It would have been interesting to compare the different pooled mortality rates of the same population measured with each DIC score. As above, comparing different populations does not provide relevant information on DIC-scores performance.

Response: Thank you for the comments. A separate meta-analysis was conducted after this comment to see the pooled mortality rate among studies that involved all the three scores at the same time. 

- There are two Figure 4. The second one provides the most relevant analysis of the manuscript. However, due to the very high heterogeneity in all sections (A-B-C) caution must be used when interpreting the results.

Response: Thank you for the comments. 

My suggestion is to rethink the analyses using a comparable population to evaluate the performance of each DIC-score.

Response: Thank you for the suggestion. Further analysis on comparable population was performed based on the suggestion

Reviewer #4: Thank you for the opportunity to review that meta-analysis about different diagnostic criteria for DIC for prognostics in sepsis patients. Due to the high mortality in septic patients, it is important to have different prognostic tools to identify critically ill patients early. This is especially important in settings with limited resources as ICU beds are scarce. In my opinion the authors performed an overall well-conducted review, were only a few questions are open, before publishing it:

- According to your manuscript, you performed a search update this August. Did you include any new studies and can you add that information in the PRISMA flow-chart (how many have you found and how many did you include). Did you include only papers in English or also other languages?

Response: Thank you for the important points. Yes, two studies were included in the later updated search and they are included in the PRISMA flow-chart. Papers in English language were included in this review.

- Can you state the inclusion and exclusion criteria based on the SPIDER tool?

Response: Thank you for the suggestion. For prediction performance meta-analysis, PICO, possibly with addition of timing and setting components (PICOTS) is more appropriate tool. The listed eligibility criteria in this review captured these criteria.

- In the abstract you say, you performed a random-effect meta-analysis, but I did not find anything about it in the full text. Please explain, why you used this method

Response: Thank you for the comments. After this comment, we included the statement in the data analysis part in the full text. The assumptions in the random-effect model best fits the current meta-analysis allowing for heterogeneity between studies.

- Two times, you state you want to evaluate the diagnosis of DIC or identify patients with DIC. Can you change this, as you only determined the score positivity rate / assessed the proportion of DIC and coagulopathy in sepsis patients

Response: Thank you for the comment. Correction has been made in the conclusion part accordingly.

- In the results section you wirte „Three studies …“ but only two studies are cited.

Response: Thank you for the comment. We corrected the error on the same line

- Please write out abbreviations first time using them (also if you write them out in the abstract, you have to write it out the first time in the full manuscript as well)

Response: Thank you for the comment. We addressed the comment throughout the text

- Please add the newest sepsis guideline: DOI: 10.1097/CCM.0000000000005337

Response: Thank you for suggesting surviving sepsis campaign international guideline for management of sepsis. There is no need to cite the reference at this time.

- Can you add one or two sentences to the pathomechanism of DIC and are there any risk factors associated with the development of DIC

Response: Thank you for the valuable comment. We added sentences regarding the mechanism of DIC in sepsis at the 1st paragraph of introduction.

- The department of the first study is missing. If it is not available, please indicate as well.

Response: Thank you for the comment. We now added the department.

- Please also consider including this recently published review: DOI: 10.7759/cureus.67052

Response: Thank you for suggesting this recent review. There is no need for now to use this particular reference.

- Could you please add the references for the following sentence „Though there are dedicated clinical scores that …“

Response: Thank you for the comment. We added the citation for the indicated statement.

- Can you compare the sensitivity and specificity to predict the mortality in patients with sepsis to the dedicated clinical scores

Response: Thank you for the comment and suggestion. Considering the performance of these clinical scores is beyond the scope of this review.

---

## [Decision Letter · Decision Letter 1]

2 Dec 2024

A comparison of Disseminated Intravascular Coagulation Scoring Systems and Their Performance to Predict Mortality in Sepsis patients: A Systematic Review and Meta-analysis

PONE-D-24-39032R1

Dear Dr. Kiya,

We’re pleased to inform you that your manuscript has been judged scientifically suitable for publication and will be formally accepted for publication once it meets all outstanding technical requirements.

Kind regards,

Mehmet Baysal

Academic Editor

PLOS ONE

Additional Editor Comments (optional):

I believe the authors adequately addressed reviewers concerns through additional subgroup analyses and explanations provided in their revised manuscript. Specifically: The authors acknowledged the high heterogeneity in their results and discussed this as a limitation. They performed subgroup analyses for studies that used all three scores within the same population, which provided additional robustness to their findings. Although only a subset of studies directly compared all three DIC scores in the same cohort, the authors performed sensitivity analyses on this subset, showing consistent patterns with the broader results. This mitigated some of the concerns regarding non-comparable populations

Reviewers' comments:

Reviewer's Responses to Questions

**Comments to the Author**

1. If the authors have adequately addressed your comments raised in a previous round of review and you feel that this manuscript is now acceptable for publication, you may indicate that here to bypass the “Comments to the Author” section, enter your conflict of interest statement in the “Confidential to Editor” section, and submit your "Accept" recommendation.

Reviewer #1: All comments have been addressed

Reviewer #3: All comments have been addressed

Reviewer #4: All comments have been addressed

2. Is the manuscript technically sound, and do the data support the conclusions?

Reviewer #1: Yes

Reviewer #3: Partly

Reviewer #4: Yes

3. Has the statistical analysis been performed appropriately and rigorously? 

Reviewer #1: Yes

Reviewer #3: No

Reviewer #4: Yes

4. Have the authors made all data underlying the findings in their manuscript fully available?

Reviewer #1: Yes

Reviewer #3: Yes

Reviewer #4: Yes

5. Is the manuscript presented in an intelligible fashion and written in standard English?

Reviewer #1: Yes

Reviewer #3: Yes

Reviewer #4: Yes

6. Review Comments to the Author

Reviewer #1: The authors have carefully evaluated reviewer comments and I believe reviewer comments have been adequately addressed.

Reviewer #3: Thank you to the authors for considering my suggestions. I do not have other comments for improving the manuscript.

Reviewer #4: Dear Authors,

thank you for considering the comments made. I think after the revisions made based on these comments this manuscript about DIC scores for patients with sepsis to predict mortality became even better. There is only one stylistic preference from me I would like to add. I would prefer "Table S1" instead of "S1 Table".

7. PLOS authors have the option to publish the peer review history of their article (what does this mean?). If published, this will include your full peer review and any attached files.

Reviewer #1: No

Reviewer #3: No

Reviewer #4: No

---

## [Editor Report · Acceptance letter]

4 Dec 2024

PONE-D-24-39032R1 

PLOS ONE

Dear Dr. Kiya, 

I'm pleased to inform you that your manuscript has been deemed suitable for publication in PLOS ONE. Congratulations! Your manuscript is now being handed over to our production team.

Kind regards, 

on behalf of

Dr. Mehmet Baysal 

Academic Editor

PLOS ONE